# Anomaly Segmentation Network Using Self-Supervised Learning

**Jou Won Song[1*], Kyeongbo Kong[2*], Ye-In Park[1], Seong-Gyun Kim[3], and Suk-Ju Kang[1]**

Department of Electronic Engineering, Sogang University, Seoul, Korea, [1]
Department of Media communication, Pukyong National University, Busan, Korea[2],
LG Display, Seoul, South Korea[3]

## Abstract

Anomaly segmentation, which localizes defective areas, is an important component in large-scale industrial manufacturing. However, most recent researches have focused on anomaly detection. This paper proposes a novel anomaly segmentation network (AnoSeg) that can directly generate an accurate anomaly map using self-supervised learning. For highly accurate anomaly segmentation, the proposed AnoSeg considers three novel techniques: Anomaly data generation based on hard augmentation, self-supervised learning with pixel-wise and adversarial losses, and coordinate channel concatenation. First, to generate synthetic anomaly images and reference masks for normal data, the proposed method uses hard augmentation to change the normal sample distribution. Then, the proposed AnoSeg is trained in a self-supervised learning manner from the synthetic anomaly data and normal data. Finally, the coordinate channel, which represents the pixel location information, is concatenated to an input of AnoSeg to consider the positional relationship of each pixel in the image. The estimated anomaly map can also be utilized to improve the performance of anomaly detection. Our experiments show that the proposed method outperforms the state-of-the-art anomaly detection and anomaly segmentation methods for the MVTec AD dataset. In addition, we compared the proposed method with the existing methods through the intersection over union (IoU) metric commonly used in segmentation tasks and demonstrated the superiority of our method for anomaly segmentation.

## Introduction

Anomaly segmentation is the process that localizes anomaly regions. In the real world, since the number of anomaly data is very limited, conventional anomaly segmentation methods are trained using only normal data. Typically, many anomaly segmentation methods are based on anomaly detection techniques (using reconstruction loss (An and Cho 2015; Baur et al. 2018; Sakurada and Yairi 2014; Chen et al. 2017; Bergmann et al. 2019), high-level learned representation (Bergmann et al. 2020) and GradCAM (Venkataramanan et al. 2020; Kimura et al. 2020)) because the real dataset includes few anomaly images without the ground truth (GT) mask. Therefore, as shown in Figs. 1(b) and (c),

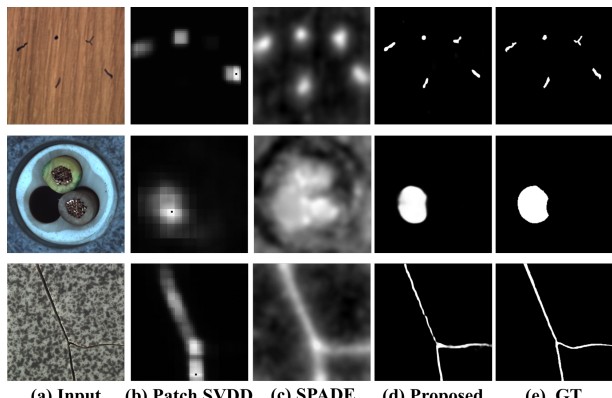

(a) Input   (b) Patch SVDD   (c) SPADE   (d) Proposed   (e) GT

Figure 1: Comparison of anomaly maps (before thresholding) of the proposed method with the SOTA methods in the MVTec-AD dataset. Except for the proposed method, anomaly maps of existing methods are normalized to [0, 1].

these methods such as Patch SVDD (Yi and Yoon 2020) and SPADE (Cohen and Hoshen 2020) are not trained directly on pixel-level segmentation and they are difficult to generate anomaly maps similar to GT masks.

To handle this problem, this paper proposes a new methodology that can directly learn the segmentation task. The proposed anomaly segmentation network (AnoSeg) can generate an anomaly map to segment the anomaly region that is unrelated to the normal class. The goal of AnoSeg is to generate an anomaly map that represents the normal class region within a given image for anomaly segmentation, unlike the existing methods to indirectly extract anomaly maps. For this goal, our AnoSeg proposes three following approaches. First, as shown in Fig. 2, AnoSeg uses the segmentation loss directly using the synthesized data generated through hard augmentation. Second, AnoSeg learns to generate the anomaly map and reconstruct normal images. Also, an adversarial loss is applied by using a generated anomaly map and an input image. Since the anomaly map learns the normal sample distribution, AnoSeg has high generalization for unseen normal and anomaly regions even with a small number of normal samples. Third, we propose the coordinate channel concatenation using a coordinate vector based

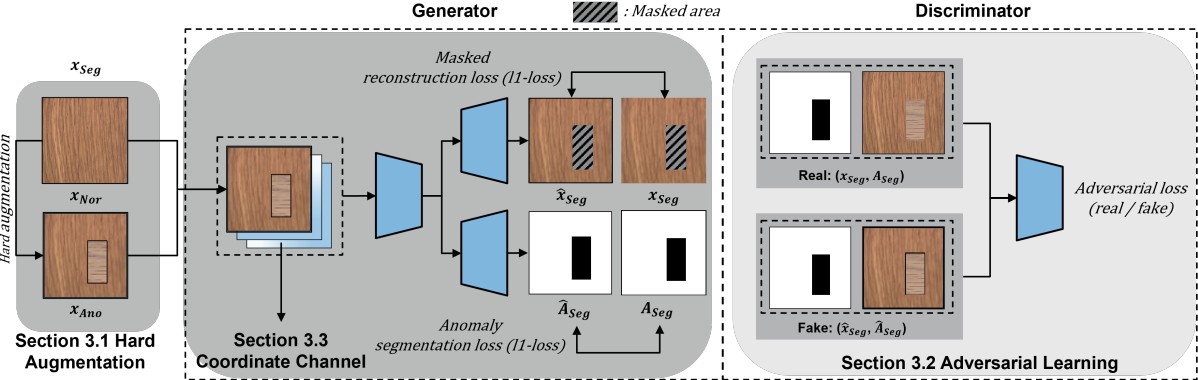

Figure 2: Overview of the training process of the proposed AnoSeg. AnoSeg generates reconstructed images and anomaly maps.

on coordconv (Liu et al. 2018). Anomaly regions in a particular category often depend on the location information of a given image. Therefore, the proposed coordinate vector helps to understand the positional relationship of normal and anomaly regions in the input image.

Therefore, Fig. 1(d) shows that the anomaly map of AnoSeg is very similar to GT even without thresholding. Moreover, we describe how to perform anomaly detection by extending existing methods (Sabokrou et al. 2018) using anomaly maps. As a result, AnoSeg outperforms SOTA methods on the MVTec AD dataset in terms of intersection over union (IoU) and AUROC. Additional experiments using IoU metric also show that AnoSeg is robust for thresholding.

## Proposed Method: AnoSeg

The proposed AnoSeg is a "holistic" approach which incorporates three techniques: self-supervised learning using hard augmentation, adversarial learning, and coordinate channel concatenation. The details are explained in the following sub-sections.

### Self-supervised Learning Using Hard Augmentation

To train anomaly segmentation directly, an image with an anomaly region and its corresponding GT mask corresponding to the image are required. However, it is difficult to obtain these images and GT masks in the real case. Therefore, the proposed method uses hard augmentation (Tack et al. 2020) and Cutpaste (Li et al. 2021) to generate synthetic anomaly data and GT masks. Hard augmentation refers to generating samples shifted away from the original sample distribution. As confirmed in (Tack et al. 2020), the hard augmented samples can be used as a negative samples. Therefore, as shown in Fig. 3, we use three types of hard augmentation: rotation, perm, and color jitter. Each augmentation is applied with a 50% chance. Then, like Cutpaste (Li et al. 2021), the augmented data is pasted into a random region of normal data to generate the synthetic anomaly data and corresponding masks for segmentation. Finally, the

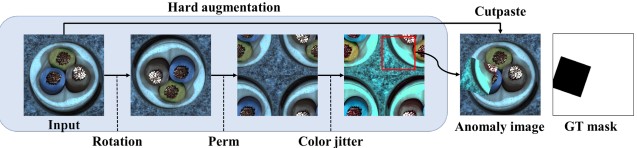

Figure 3: Our synthetic anomaly data augmentation. The synthetic anomaly data is generated by several hard augmentations and Cutpaste ((Li et al. 2021)).

anomaly segmentation dataset is composed as follows:

$$x_{Seg} = \{x_{Nor}, x_{Ano}\}, A_{Seg} = \{A_{Nor}, A_{Ano}\}, \quad (1)$$

where $x_{seg}$ is a set of normal and synthetic anomaly images, in which $x_{Nor}$ and $x_{Ano}$ are normal images and synthetic anomaly images, respectively. $A_{seg}$ is a set of normal and synthetic anomaly masks, in which $A_{Nor}$ and $A_{Ano}$ are normal masks with all inner values set to one and synthetic anomaly masks, respectively.

Using the anomaly segmentation dataset with a pixel-level loss, we can directly train our AnoSeg. The anomaly segmentation loss $L_{Seg}$ is as follows:

$$L_{Seg} = \mathbb{E} \parallel A_{Seg} - \widehat{A}_{Seg} \parallel^1, \quad (2)$$

where $\widehat{A}_{Seg}$ indicates the generated anomaly map (normal and anomaly classes). The generated anomaly map has the same size as an input image and outputs a value in the range of [0, 1] for each pixel depending on the importance of the pixel of the input image. However, since the synthetic anomaly data are only subset of various anomaly data, it is difficult to generate a real anomaly maps that are unseen in training phase.

### Adversarial Learning with Reconstruction

To improve the generality for various anomaly data, it is important to train normal region distribution accurately. Therefore, AnoSeg utilizes masked reconstruction loss that uses reconstruction loss only in normal regions to learn only the distribution of normal regions and avoid bias of the distribution of synthetic anomaly regions. Also, since the discriminator inputs a pair for an input image and its GT masks,

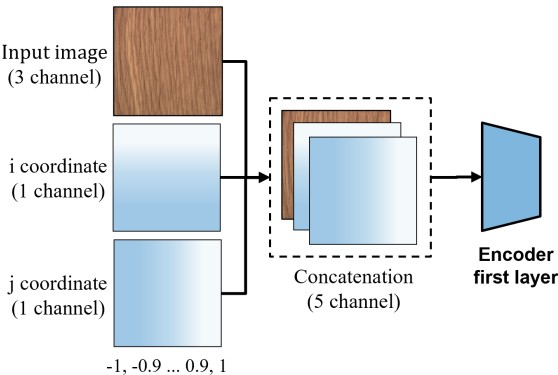

Figure 4: Overall process of the coordinate channel concatenation.

the discriminator and generator can focus on normal region distribution. Thus, anomaly region cannot be reconstructed well and the detail of the anomaly map can also be improved. Loss functions for adversarial learning are as follows:

$$L_{Adv} = \min_{G}\max_{D}\{\mathbb{E}\left[\log(D(concat(x_{Seg}, A_{Seg})))\right]$$
$$+ \mathbb{E}\left[\log(1 - D(concat(\widehat{x}_{Seg}, \widehat{A}_{Seg})))\right]\}, \quad (3)$$

$$L_{Re} = \mathbb{E}\parallel x_{Seg} * A_{Seg} - \widehat{x}_{Seg} * A_{Seg} \parallel^1 / \mathbb{E}\parallel A_{Seg} \parallel^1, \quad (4)$$

where $D$, $G$, and $concat$ are a discriminator, a generator, and a concatenation operation, respectively. In Section 5, we demonstrated the effectiveness of adversarial loss.

## Coordinate Channel Concatenation

In the anomaly segmentation task, unlike typical segmentation task, the location information is the most important information because normal and anomaly class can be changed depending on where they are located even for the same object (e.g. cable). To provide additional location information, we use a coordinate vector inspired by Coord-Conv (Liu et al. 2018). We generate rank 1 matrices that are normalized to [-1, 1] for x and y axes, respectively, considering the coordinates. Then, we concatenate these matrices with the input image as channels (Fig. 4). As a result, AnoSeg extracts features by considering the positional relationship of the input image.

## Anomaly Detection Using Proposed Anomaly Map

In this section, we design a simple anomaly detector that adds the proposed anomaly map to the existing GAN-based detection method (Sabokrou et al. 2018). The proposed anomaly detector trains anomaly detection by learning only normal data distribution after training of AnoSeg. As shown in Fig. 5, We simply concatenate the input image and anomaly map to use them as inputs of detector, and apply both an adversarial loss and a reconstruction loss. Then, we use the feature matching loss introduced in (Salimans et al. 2016) to stabilize the learning of the discriminator.

In the test process (Fig. 5), the proposed anomaly detector obtains anomaly scores using the discriminator that has

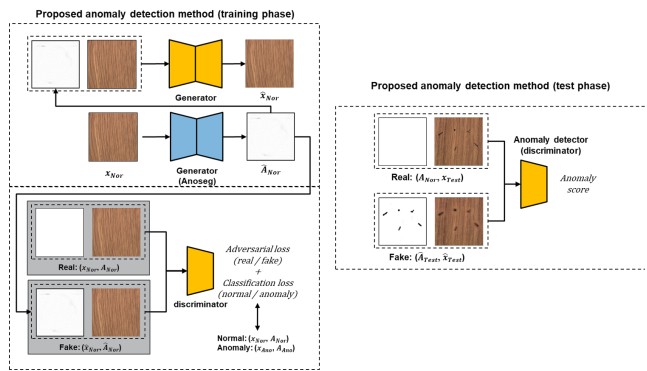

Figure 5: An overview of the proposed anomaly detection method. To obtain anomaly score, the pair of images reconstructed from the anomaly map and the anomaly detector (fake pair) are compared with the pair of the normal mask and the input image (real pair) using a discriminator.

learned the normal data distribution. We first assume that the input image is normal, so the mask $A_{Nor}$ with all inner values set to zero is used in pairs with the input image. When the input image is really normal, a fake pair (anomaly map and reconstructed image) is similar to the real pair (normal mask and input image), so the anomaly detector has a low anomaly score. On the other hand, when the input image is abnormal, the fake pair is significantly different to the real pair, so it has a high anomaly score. To compare the real and fake pair, the reconstruction loss and the feature matching loss are used as follows:

$$Score = \alpha L(f(concat(x_{Test}, A_{Nor})), f(concat(\widehat{x}_{Test}, \widehat{A}_{Test})))$$
$$+ \beta L(x_{Test}, \widehat{x}_{Test}), \quad (5)$$

where $A_{Nor}$ is a normal GT mask (all zero), $L$ is the mean squared error, $x_{Test}$ is test input image. $\widehat{A}_{Test}$ and $\widehat{x}_{Test}$ represent generated anomaly map and reconstructed image.

## Experimental Results

### Datasets and Metrics

To verify the anomaly segmentation and detection performance of the proposed method, several evaluations were performed on the MVTec AD dataset (Bergmann et al. 2019). For the MVTec AD dataset, we resized both training and testing images to the size of 256 × 256. We adopted the pixel-level and image-level AUROCs to quantitatively evaluate the performance of different methods for anomaly segmentation and detection, respectively. In addition, we used IoU to evaluate anomaly segmentation. For the measurement of IoU, a threshold, which maximizes IoU, was applied in each method.

### Implementation Details

The encoder of AnoSeg consists of the convolution layers of ResNet-18 (He et al. 2016). The up-sampling layer of decoders consists of one transposed convolution layer and

| | Pixel-level AUROC / IoU / Image-level AUROC | | | | | | |
|--------|--------|--------|--------|--------|--------|--------|--------|
| Method | CAVGA | US | FCDD | Patch$_{SVDD}$ | SPADE | Cutpaste | Proposed |
| Mean | 0.89/0.47/0.82 | 0.88/0.24/0.84 | 0.92/ - / - | 0.96/0.43/0.92 | 0.96/0.48/0.86 | 0.96/ - /0.95 | **0.97/0.54/0.96** |

Table 1: Performance comparison of anomaly segmentation and detection in terms of pixel-level AUROC and image-level AUROC with the proposed method and conventional SOTA methods on the MVTec AD dataset (Bergmann et al. 2019). Full results for anomaly detection are added in Tables 2 and 3 of Appendix A

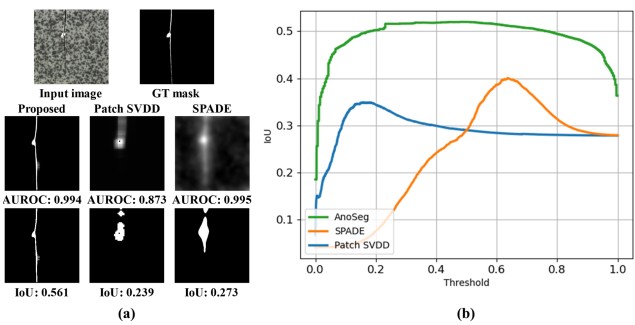

Figure 6: Overall process of the coordinate channel concatenation.

convolution layers. Two decoders of the AnoSeg are composed of five up-sampling layers and two convolution layer to generate an anomaly map and a reconstructed image. The structure of the anomaly detector is the same as the AnoSeg structure except for the decoder that generates the anomaly map. Detailed information on training process and the network architecture is described in Appendix B.

## Comparison with the state-of-the-art methods

The comparison evaluations were performed with 6 recent deep learning-based methods, both GradCAM-based methods (CAVGA (Venkataramanan et al. 2020) and Cutpaste (Li et al. 2021)) and high-level feature representation-based method (Uninformed students (US) (Bergmann et al. 2020) and FCDD (Liznerski et al. 2021), patch SVDD (Yi and Yoon 2020), and SPADE (Cohen and Hoshen 2020)) as benchmarks. AnoSeg is trained directly on segmentation, unlike conventional methods that depend on loss unrelated to the segmentation task, such as classification loss. As a result, as shown in Table 1, AnoSeg outperformed the conventional SOTA, such as Patch SVDD, SPADE, and Cutpaste, by 1% AUROC in anomaly segmentation. We also evaluated IoU, which is typically used as a metric for segmentation. Table 1 shows the quantitative comparison on IoU. AnoSeg achieved the highest performance compared to other methods in IoU. In particular, Patch SVDD and SPADE achieved 0.96 AUROC similar to AnoSeg in the evaluation of AUROC, but had lower IoU than the proposed method. This is because, unlike the existing method, the proposed method was directly trained for segmentation.

Additionally, we compared the AUROC and IoU metrics for the generated anomaly map in Fig. 6. In general, AUROC is affected by the detection performance of the anomaly regions. False positives for normal regions have relatively no impact on AUROC. In the Patch SVDD of Fig. 6(a), there were abnormal regions that cannot be detected. Therefore, the anomaly map of Patch SVDD had lower AUROC compared to other methods. Although the anomaly maps of AnoSeg and SPADE visually show different anomaly maps, the same AUROC was calculated because most anomaly regions are detected in anomaly maps of AnoSeg and SPADE. However, IoU was affected by false positives in normal regions. Therefore, IoU of SPADE had lower performance compared to AUROC. The proposed AnoSeg achieved the highest performance for both IoU and AUROC.

In addition, Patch SVDD and our AnoSeg were compared to verify the performance variation depending on the threshold of the proposed method. IoU was measured by dividing the anomaly score by 10000 units. Fig. 6(b) shows the mean performance change of AnoSeg, SPADE and Patch SVDD according to a threshold. As shown in Fig. 6(b), the performance of AnoSeg did not significantly change significantly for different thresholds. Therefore, the anomaly map is shown similar to the GT mask even though thresholding was not applied in Fig. 6(a). On the other hand, Fig. 6(b) shows that Patch SVDD and SPADE had a significant change in performance when the threshold is changed around the threshold with the highest IoU. The result shows that our model is robust against thresholding. By setting the threshold between 0.2 and 0.8, AnoSeg could always achieve better results consistently than other SOTA solutions listed in Table 1.

## Conclusion

This paper presented a novel anomaly segmentation network to directly generate an anomaly map. We proposed AnoSeg, a segmentation model using adversarial learning, and the proposed AnoSeg was directly trained for anomaly segmentation using synthetic anomaly data generated through hard augmentation. In addition, anomaly regions sensitive to positional relationships were more easily detected through coordinate vectors representing the pixel position information. Hence, our approach enabled AnoSeg to be trained to generate anomaly maps with direct supervision. We also applied this anomaly maps to existing methods to improve the performance of anomaly detection. Experimental results on the MVTec AD dataset using AUROC and IoU demonstrated that the proposed method is a specialized network for anomaly segmentation compared to the existing methods.

## Acknowledgment

This research was supported by the MSIT(Ministry of Science and ICT), Korea, under the ITRC(Information Technology Research Center) support program(IITP-2021-2018-0-01421) supervised by the IITP(Institute of Information communications Technology Planning Evaluation), and was supported by the National Research Foundation of Korea (NRF) grant funded by the Korea government (MSIT) (No. 2021R1A2C1004208).

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
