# OpenReview forum: "Anomaly Segmentation Network Using Self-Supervised Learning"
_AAAI.org/2022/Workshop/ADAM — AAAI 2022 Workshop ADAM_

### Official Review · Reviewer_cNDN · 2021-11-29
**Conceptually clear paper with convincing results. Additional details will make paper easier to follow**

**Rating:** 7
**Confidence:** 3

**Review:**

This paper discusses a method for anomaly segmentation using ideas from multiple fields, including hard augmentation to generate artificial data with anomalies, adversarial training for improved generalization, and coordinate channel concatenation to learn positional features. The authors compared their results to several state-of-the-art methods on a benchmark dataset and demonstrated that their methods outperformed other baselines and were significantly more robust to various thresholding values. The overall concept is clearly presented at a high level and the results shown are also convincing and would be significant, since the challenge of obtaining sparse anomalous data, can be circumvented. A couple of things which can be improved upon are:

The section on coordinate concatenation is not very clear and Figure 4 is not very informative as well. Additional descriptions on this section (maybe in the appendix) will make the paper easier to follow

It is also not clear to me what the discriminator during the training phase and the anomaly detector during the testing phase takes as input. During the testing phase, how is the fake data generated? Is it using the generator? Overall, a more descriptive section detailing the flow of data during training and testing will be helpful.

---

### Official Review · Reviewer_EeVd · 2021-12-02
**Solid paper**

**Rating:** 7
**Confidence:** 4

**Review:**

The paper proposes a method for anomaly segmentation - a problem widely observed in manufacturing. For that purpose, they leverage hard augmentation, self-supervised learning for generation, and a discriminator for anomaly detection. Anoseg provides promising results when compared to existing methods.

Overall, the paper is clearly written. However, the full end-to-end flow of the pipeline is hard to follow from the figures and the main text. I personally found the abstract provides a clear overview of the entire flow. I would suggest authors to include/reiterate the training/testing flow in the main text, as well as to expand figure captions,  to make the paper easier to follow.

Authors should include recent relevant works on novelty detection/generation/anomaly segmentation using deep generative models,  for example (1) "RaPP: Novelty Detection with Reconstruction along Projection Pathway", ICLR 2020; (2) "Toward A Neuro-inspired Creative Decoder", IJCAI 2020; (3)"DFR: Deep Feature Reconstruction for Unsupervised Anomaly Segmentation", arXiv:2012.07122.